# The association between maternal intra-abdominal pressure and hypertension in pregnancy

**Sajith Jayasundara**[1]*, **Malik Goonewardene**[1,2], **Lanka Dassanayake**[1,2]

1 Academic Obstetric Unit, Teaching Hospital, Galle, Sri Lanka, 2 Department of obstetrics and gynaecology, Faculty of Medicine, University of Ruhuna, Galle, Sri Lanka

* sajithjayasundara@yahoo.co.uk

## Abstract

### Introduction

Pregnancy leads to a state of chronically increased intra-abdominal pressure (IAP) caused by a growing fetus, fluid, and tissue. Increased intra-abdominal pressure is leading to state of Intra-Abdominal Hypertension (IAH) and Abdominal Compartment Syndrome. Clinical features and risk factors of preeclampsia is comparable to abdominal compartment syndrome. IAP may be associated with the hypertension in pregnancy (HIP).

### Objectives

The study aimed to determine the antepartum and postpartum IAP levels in women undergoing caesarean delivery (CD) and association between hypertension in pregnancy, and antepartum and postpartum IAP levels in women undergoing CD.

### Method

Seventy pregnant women (55 normotensive, 15 HIP) undergoing antepartum, non-emergency CD, had their intravesical pressure measured before and after the CD, the intravesical pressure measurements obtained with the patient in the supine position were considered to correspond to the IAP. Multivariable linear regression models were used to study associations between intraabdominal pressure and baseline characteristics in normotensive pregnancies and hypertensive pregnancies.

### Results

In normotensive pregnancies at mean gestation age of 38.2 weeks (95%CI 37.9 to 38.6), mean antepartum IAP was 12.7 mmHg(95%CI 11.6 to 13.8) and the mean postpartum IAP was 7.3 mmHg (95% CI 11.6 to 13.8). Multivariable linear regression models showed HIP group antepartum IAP positively associated with coefficient value of 1.617 (p = 0.268) comparing with normotensive pregnancy group. Postpartum IAP in HIP group positively associated with coefficient value of 2.519 (p = 0.018) comparing with normotensive pregnancy group. IAP difference is negatively associated with HIP (coefficient -1.013, p = 0.179).

**Data Availability Statement:** All relevant data are within the manuscript.

**Funding:** The authors received no specific funding for this work.

**Competing interests:** The authors have declared that no competing interests exist.

## Conclusion

In normotensive pregnancies at term, the IAP was in the IAH range of the non-pregnant population. Higher Antepartum IAP and Postpartum IAP are associated with HIP. Reduction of IAP from antepartum period to postpartum period was less with HIP.

## Introduction

Pregnancy with growing fetus and amniotic fluid, leading to increased pressure within the abdominal cavity. The importance of the diagnosis and management of IAH and abdominal compartment syndrome is increasingly recognized in different fields of medicine. To date, little is known about normal values of IAP during pregnancy either in healthy or complicated pregnancies.

Despite decades of research into the etiology and mechanism of preeclampsia, its exact pathogenesis remains uncertain. Several authors have hypothesized that IAP as an etiologic factor of HIP. Risk factors and clinical manifestation of abdominal compartment syndrome are similar to the preeclampsia and eclampsia. Supporting this hypothesis is the fact that HIP is more often seen in association with a first pregnancy (when the abdomen has not been previously stretched) than subsequent pregnancies, with twin pregnancies which would be more likely to be associated with an increased IAP and with obesity [1]. An animal study has shown that chronically increased intra-abdominal pressure would lead to systemic hypertension [2]. In the 1900s, Paramore had suggested uncompensated elevated IAP as a possible etiologic factor in the development of preeclampsia [3].

Harvey Sugerman hypothesized that preeclampsia is a venous disease secondary to an increased intra-abdominal pressure. Pregnancy with high IAP may produce all of the pathology associated with preeclampsia secondary to compresses the inferior vena cava, uterine veins, portal vein, hepatic veins, splenic vein and renal veins which lead to a decreased flow in these vascular beds [4]. It is postulated that this decreased venous flow increases progressively throughout the body as the IAP rises, leading to lower body edema, placental ischemia and infarction, decreased renal venous flow leading to activation of the juxtaglomerular apparatus, hypertension and proteinuria, decreased portal and splenic venous flow with hepatic ischemia, necrosis, elevated liver enzymes (transaminases), hypersplenism with thrombocytopenia and hemolysis, increased intra-thoracic pressure with upper body edema, decreased jugular venous flow, increased intra-cranial pressure, and seizures. According to the presumed pathophysiology, the association between elevated intra-abdominal pressure and HIP is biologically plausible. However, the contribution of increased IAP to the pathogenesis of HIP may be influenced by other individual factors. The development of HIP is influenced by multiple factors.

The placenta plays central role in the pathogenesis of preeclampsia, specifically in relation to the imbalance of placental pro-angiogenic and anti-angiogenic factors. Genetic factors and immune factors have also been established as contributors to the development of preeclampsia. Additionally, increased IAP may serve as a contributing factor [5]. The extent of each factor's contribution to the pathogenesis may vary depending on individual circumstances. After delivery of fetus and placenta, IAP levels typically decrease. Nevertheless, the effects of pathological processes following delivery can either linger or diminish, contingent upon individual factors. In particular, the presence of elevated levels of placental pathological agents in circulation post-delivery can contribute to the development of postpartum preeclampsia and hypertension [6].

The role of IAP in the development of preeclampsia may align with the traditional concept of disease causation, where the virulence of the agent and the susceptibility of the host play a crucial role. Consequently, increased IAP might not always lead to preeclampsia in situations that raise the IAP, for instance, polyhydramnios, fetal macrosomia, uterine fibroids, or ovarian cysts.

In the pathogenesis of preeclampsia, the presence of amplified anti-angiogenic factors from the placenta is believed to be particularly prominent in cases of molar pregnancy as compared to other factors. Additionally, changing partners between pregnancies increases the risk of preeclampsia, where immune maladaptation at the fetal-maternal interface could be an underlying mechanism more than other factors.

Despite being proposed by several authors, there is currently a lack of research evidence to support the hypothesis that IAH is an etiological factor of gestational hypertension. This study aims to generate further evidence to fill this gap in the knowledge base regarding this hypothesis.

The objective of our study was to assess the levels of IAP in both normotensive pregnancies and pregnancies with hypertension, as well as to determine the relationship between IAP and hypertension during pregnancy.

## Method

### Ethical considerations

Ethical approval was obtained from Ethical Review Committee, Faculty of Medicine, University of Ruhuna, Galle, Sri Lanka. Informed written consent was obtained from all the participants in the study. The enrolment was carried out in 2016 and 2017.

### Study design and setting

A cross sectional study was carried out on a cohort of 70 women with singleton pregnancies, undergoing antepartum, non-emergency CD in the Academic Obstetric Unit, Teaching Hospital, Mahamodara, Galle, Sri Lanka. Pregnant women underwent Category 1 CD and CD performed during active phase of labour, are excluded.

Mothers with vaginal deliveries were not included as urinary catheterisation, which is needed for measuring IAP, is not a routine procedure among mothers undergoing vaginal deliveries.

Antenatal care data was collected to data sheet, from in-ward clinical records, antenatal care records and investigation results.

### Intra-abdominal pressure measurement

The participants were transferred to the operating room and the pre-operative blood pressure was recorded. Prior to spinal or general anaesthesia, patients were placed in the supine position, and a transurethral 16-Fr Foley catheter was inserted into the bladder with non-touch technique under aseptic conditions. Urinary bladder was emptied and the sample of urine checked for albumin. Intra vesical pressure measurements was taken using the Intravesical pressure measurement system according to the world abdominal compartment society recommendation with patient in supine position, mid axillary level as zero reference and 25ml of bladder inflation volume.

All the measurements were obtained by the principal investigator. Three consecutive measurements were obtained at the end of expiration and mean value of the three measurement was calculated. As part of the post-CD analgesic protocol, effective pain management was

ensured by consistently evaluating post-operative pain levels using the 0–10 visual analogue scale (VAS) and administering analgesics in accordance with the WHO analgesic ladder. IAP measurements were repeated between six to eight hours post-operatively. In cases where the VAS score was less than 3, IAP measurement was performed. However, if the VAS score was 3 or higher, analgesics were administered and IAP measurement was subsequently conducted after confirming a VAS score below 3. Postoperative complications, such as bleeding, infections, constipation, and ileus, were not observed in the patients during their hospital stay. Birth weight of the babies were measured using standard weight measuring scale.

## Statistical analysis

The Statistical Package for Social Sciences (SPSS 22.0; SPSS Inc., Chicago, IL) version 22.0 was used for statistical analysis. Data normality was assessed by the Kolmogorov-Smirnov test. Statistical comparisons among groups were performed using the independent $t$ test and Chi-square test. All $p$ values were two tailed and statistical significance was set as $p < 0.05$.

The effect of belonging in the hypertensive group on the dependent variables antepartum IAP, postpartum IAP and difference between antepartum IAP and postpartum IAP was estimated using least squares multivariable regression analyses, adjusting for maternal BMI, birthweight, gestational age and parity. Linear regression analysis was used as the dependent variable is continuous in all three cases (antepartum IAP, postpartum IAP, IAP difference). Firstly, a pearson correlation analysis was conducted in order to test for pairwise correlations between BMI, birthweight and gestational age. Birthweight and gestational age were found to be highly positively correlated (correlation coefficient: 0.78) therefore it was decided that birthweight would need to be retained and gestational age would need to be removed from all subsequent regression analyses.

For each of the three dependent variables, different combinations of regression models were fit in order to assess all possible combinations of variables and choose the model with the highest adjusted $R^2$ value. The adjusted $R^2$ value is a robust goodness-of-fit metric in linear regression models and shows the percentage of the variance of the dependent variable that is attributed to the independent variables, accounting for the number of variables in the model. Squared transformations of variables in the regression models were also examined in order to check for improvements in model fitting. Hypertension was included in the models as a binary categorical variable; hypertensive subjects were assigned the value 1 while normotensive the value 0. A p value lower than 0.05 corresponding to each coefficient in the model was used as a criterion for significance.

## Results

Seventy subjects who undergoing CD were enrolled. Subjects were categorized into, normotensive (n = 55, 78.6%) and HIP (n = 15, 21.4%). HIP include Gestational hypertension (n = 5), Preeclampsia (n = 8), Eclampsia (n = 1) and chronic hypertension (n = 1). Mean values with standard deviations and 95% confidence intervals or Median with interquartile range calculated in each parameter calculated in each category as shown in Table 1.

Outcome measures in normotensive category and hypertension in pregnancy category compared with Systemic blood pressures, IAP parameters, and birth weight of the neonates. Mean values with 95% Confidence interval calculated in each parameter calculated in each category as shown in Table 2.

Different combinations of variables in each model were tested in order to retain the most efficient model for each dependent variable. The most robust and parsimonious regression model, i.e. the model with the highest adjusted $R^2$ (0.032), when considering the effect of the

**Table 1. Characteristics of subjects.**

|  | Normotensive (n = 55) | Hypertension in Pregnancy (n = 15) | p value |
|---|---|---|---|
| Age (years) |  |  |  |
| Mean (95% CI) | 32.5 (31.4 to 33.7) | 32.9 (29.8 to 35.9) | 0.805* |
| GA (weeks) |  |  |  |
| Mean (95% CI) | 38.2 (37.9 to 38.6) | 34.3 (32.4 to 36.1) | 0.000* |
| Parity |  |  |  |
| Median(IQR) | 2.0(1) | 2.0 (3.0) | 0.220** |
| BMI (kg/m$^2$) |  |  |  |
| Mean (95% CI) | 23.6 (22.9 to 24.3) | 24.5 (23.1 to 25.9) | 0.241* |

CI = Confidence interval GA = Gestational age IQR = Interquartile Range

* Independent t test

** Chi-square test

aforementioned independent variables on antepartum IAP included hypertension and birthweight as independent variables. Birthweight (p = 0.042) was found to be a significant predictor of antepartum IAP while hypertension was found to be positively associated to antepartum IAP (coef: 1.617) although a statistically significant effect was not established (p = 0.268). Distribution of antepartum IAP level against the birth weight shown in Fig 1.

The most efficient model with respect to predicting IAP difference ($R^2$: 0.190) included the variables hypertension (p = 0.179) and birthweight squared (p = 0.01). Hypertension was negatively associated (coef: -1.01) to IAP difference although a statistically significant

**Table 2. Systemic blood pressure and Intra- abdominal pressure in normotensive category and hypertension in pregnancy category.**

|  | Normotensive (n = 55) | Hypertension in Pregnancy (n = 15) | p value |
|---|---|---|---|
| Birth weight (grams) |  |  |  |
| Mean | 2934.5 | 1886.7 | 0.000* |
| 95% CI | 2798.6 to 3070.5 | 1415.9 to 2357.4 |  |
| Antepartum MAP (mmHg) |  |  |  |
| Mean | 87.5 | 122.2 | 0.000* |
| 95% CI | 85.9 to 89.1 | 114.5 to 129.9 |  |
| Postpartum MAP (mmHg) |  |  |  |
| Mean | 86.0 | 101.3 | 0.000* |
| 95% CI | 84.4 to 87.7 | 94.4 to 108.2 |  |
| Antepartum IAP (mmHg) |  |  |  |
| Mean | 12.7 | 12.5 | 0.892* |
| 95% CI | 11.6 to 13.8 | 10.0 to 15.0 |  |
| Postpartum IAP (mmHg) |  |  |  |
| Mean | 7.3 | 9.2 | 0.022* |
| 95% CI | 6.5 to 8.0 | 7.3 to 11.2 |  |
| IAP difference[a] (mmHg) |  |  |  |
| Mean | 5.4 | 3.3 | 0.002* |
| 95% CI | 4.8 to 6.0 | 2.1 to 4.6 |  |

95%CI = 95% Confidence interval, MAP = Mean arterial pressure, IAP = Intra-abdominal pressure a—IAP difference = Pre caesarean Delivery intra-abdominal pressure–Post caesarean delivery intra-abdominal pressure

* Independent t test

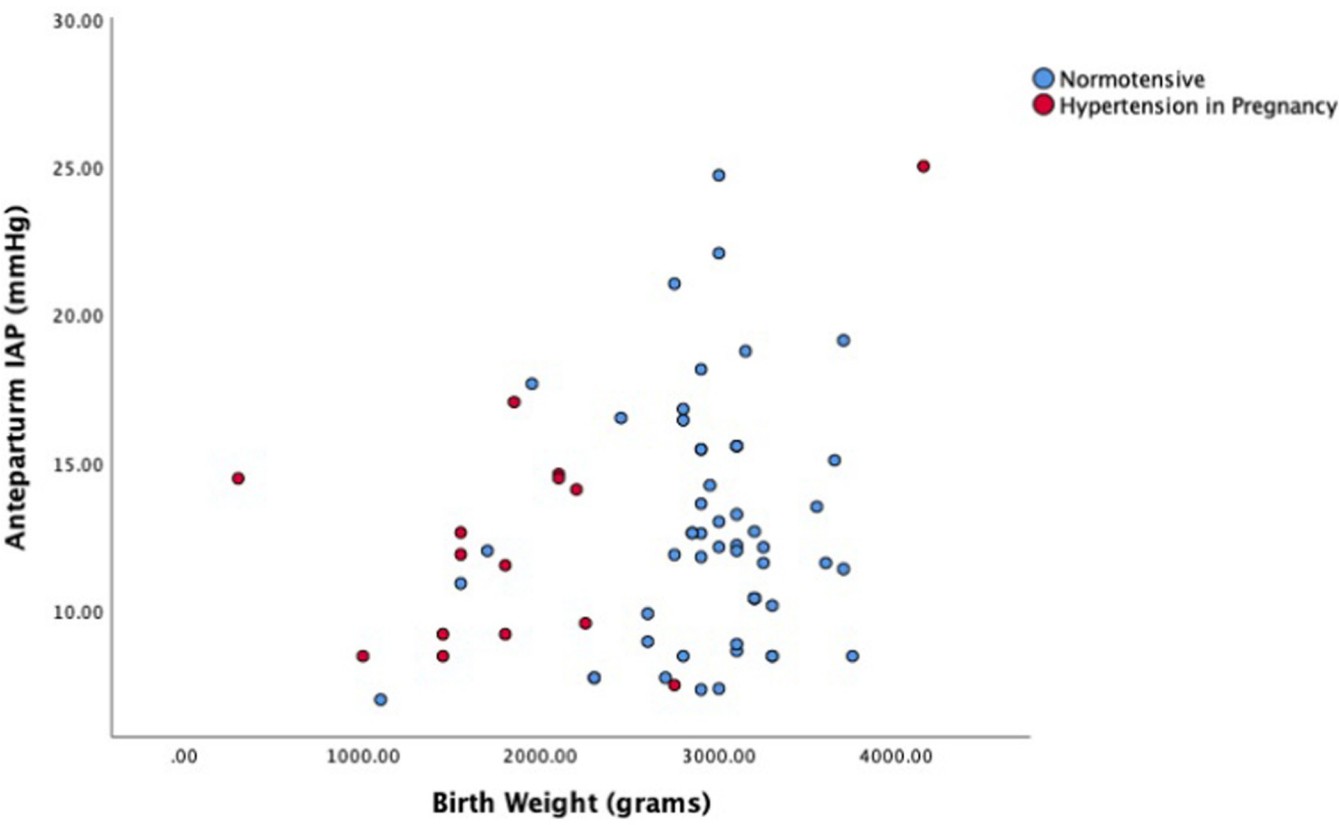

**Fig 1. Antepartum IAP level and the birth weight in normotensive and hypertension in pregnancy.**

effect was not established (p = 0.179). Distribution of IAP difference level against the birth weight shown in Fig 2.

The most efficient model with respect to predicting postpartum IAP ($R^2$: 0.059) included hypertension and birthweight. Hypertension was found to be positively (coef: 2.519) and significantly associated to postpartum IAP (p = 0.018). Distribution of postpartum IAP level against the birth weight shown in Fig 3.

Overall, for all three dependent variables, hypertension and birthweight produced the best adjusted $R^2$ scores so these were the variables that were chosen to be included in the models.

## Discussion

### Intra-abdominal pressure in normotensive pregnancy

In this study, 55 of normotensive subjects at term, antepartum mean IAP was 12.7 mmHg (95% CI = 11.6 to 13.8) and postpartum mean IAP 7.3 mmHg (6.5 to 8.0). Comparing with other five studies published regarding antepartum and postpartum IAP is shown in Table 3.

In this study, normotensive pregnancies at term, antepartum mean IAP is within the range of three studies, Fuchs F et al [7],Chun R et al [8] and Staelens A et al [9]. In this study, antepartum mean IAP slightly lower than Fuchs F et al and Staelens A et al antepartum mean IAP which may attribute with birth weight and other demographic factors. Al Khan A *et al* [10] study IAP value is higher than our antepartum IAP. This higher value of IAP, probably due to methodological overestimation, as is noticed by Chun et al [8] and Staelens A. et al [9].

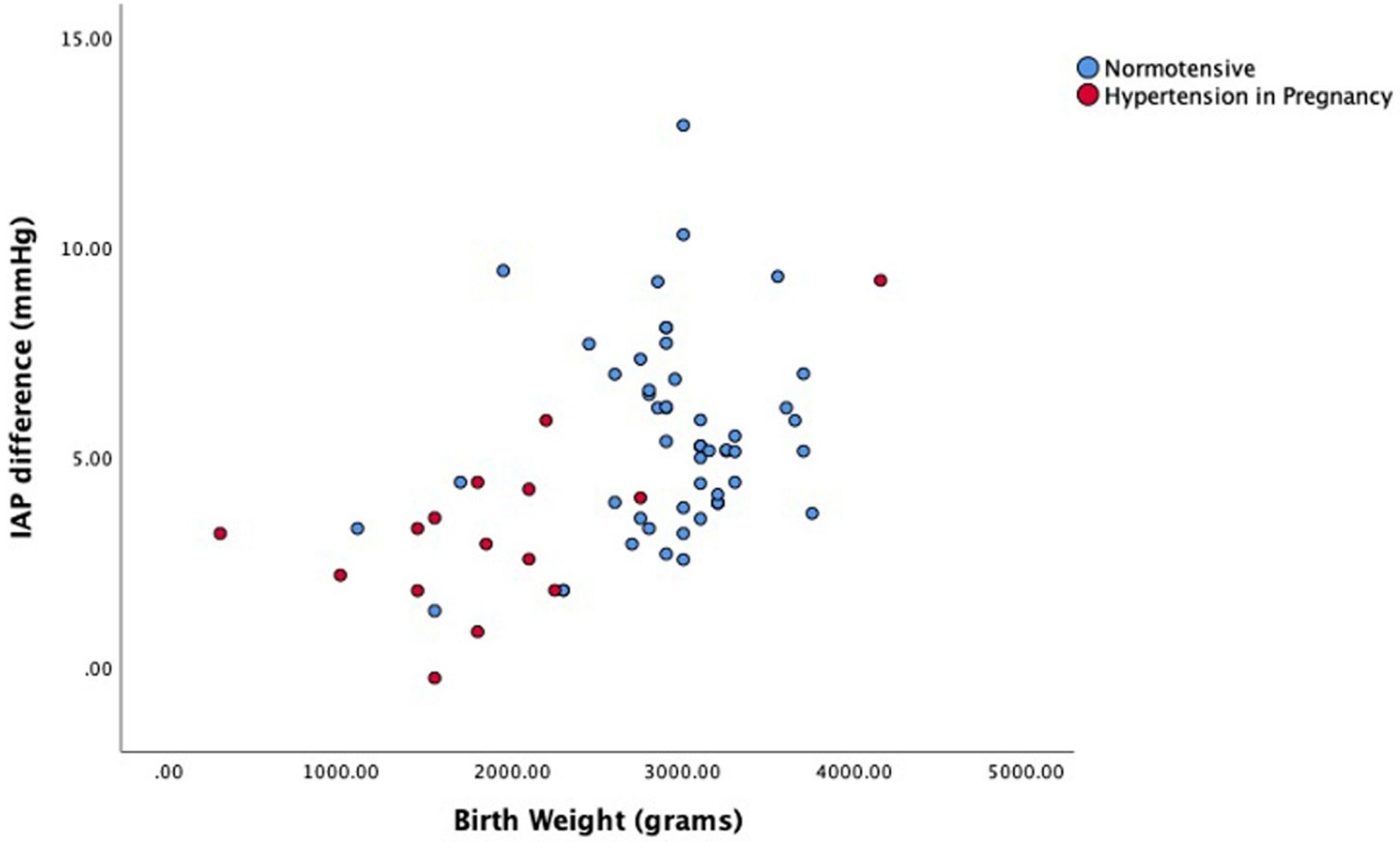

**Fig 2. IAP difference and the birth weight in normotensive and hypertension in pregnancy.**

Postpartum mean IAP in this study was within the range of Postpartum IAP of Fuchs F et al [7], Staelens A et al [9], and Abdel-Razeq et al [11] studies. Al Khan A et al [10] study IAP value is higher than our postpartum mean IAP.

According to the latest systematic review, the antepartum IAP in singleton normotensive pregnancies ranged from 7.3 to 14.1 mmHg. The postpartum IAP ranged from 3.7 to 10.8 mmHg [12].

## Intra-abdominal pressure in hypertension in pregnancy

In our study, antepartum, postpartum mean IAP levels are 12.5 mmHg (10.0 to 15.0) and 9.2 mmHg (7.3 to 11.1) respectively at mean gestational age and 34.3 weeks and mean birth weight of 1886.66g. Antepartum IAP is within in the range of Grade 1 IAH. Postpartum IAP level slightly higher than the normal level of IAP in non-pregnant population.

Mesut A. Ünsal et al and Arora V et al have compared the in IAP in pregnancies with HIP and pregnancies without hypertension [13, 14]. These studies found that the IAP in both categories, for both antepartum and postpartum periods, fell within a similar range, as shown in Table 4.

In this study, Multivariable regression analyses were used to compare the IAP levels in two categories. Because two groups have different birth weights and gestational age which directly influence the IAP levels.

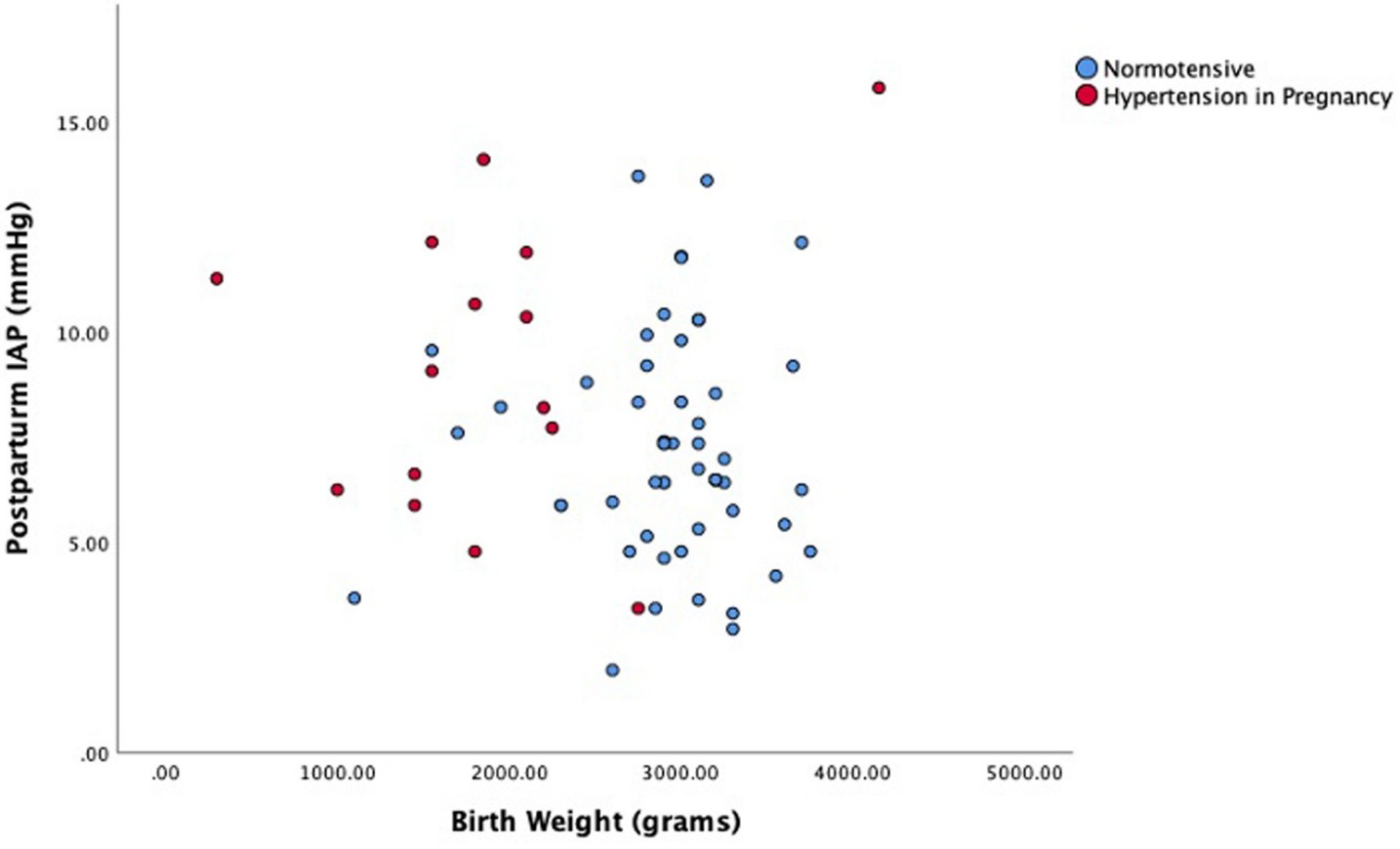

**Fig 3. Postpartum IAP and the birth weight in normotensive and hypertension in pregnancy.**

Regression analysis of antepartum IAP showed 1.6 times higher IAP level in hypertension in pregnancy compared to the normotensive group when adjusted for birthweight. However, it is not statistically significant, which is probably due to the small sample size in the

**Table 3. Comparison of studies of intra-abdominal pressure in normotensive pregnancies.**

| Author | Our study | Fuchs F *et al* 2013 [7] | Chun R *et al* 2012 [8] | Staelens A *et al* 2014 [9] | Al Khan A *et al* 2011 [10] | Abdel-Razeq *et al* 2010 [11] |
|---|---|---|---|---|---|---|
| Country | Sri Lanka | France | Canada | Belgium | USA | USA |
| Sample size | 55 | 70 | 20 | 23 | 100 | 21 |
| Age (Years) | 32.5 (31.4 to 33.7) | 34 (27–41) | 31 (SD 5) | 29.3 (SD 3.5) | 31.7 (5.1) | - |
| Gestational Age (Weeks) | 38.2 (37.9 to 38.6) | 38.1 (38 to 39) | 38.6 (SD 1.6) | 39.0 (0.6) | 38.4 (0.8) | - |
| Birth weight | 2934.5 (2799 to 3071) | 3210 (2445 to 4215) | - | 3590 (SD- 472.2) | 3422.8 (450.1) | - |
| **Antepartum IAP (mmHg)** | **12.7 (11.6 to 13.8)** | **14.2 (6.3 to 23)** | **10.9 (6.23 to 15.57)** | **14.0 (11.4 to 16.6)** | **22.0 (2.9)** | - |
| **Postpartum IAP (mmHg)** | **7.3 (6.5 to 8.0)** | **11.5 (5 to 19.7)** | - | **9.8 (6.8 to 12.8)** | **16.4 (2.6)** | **6.4 (1.2 to 11.6)** |
| Measuring technique | Supine, 25ml, level of mid axillary line | Supine, 25ml, level of symphysis pubis | Supine, 25ml, level of mid axillary line | Supine, 20ml, level of mid axillary line | Supine with leftward tilt, 50ml | Supine, 25ml, symphysis pubis |

IAP–Intra abdominal pressure, SD–Standard deviation

**Table 4. Comparison of intra-abdominal pressure in hypertension in pregnancy.**

| | Our study | Mesut A. Ünsal et al 2016 | Arora V et al 2020 |
|---|---|---|---|
| Category | Hypertension in pregnancy | Hypertension in pregnancy | Preeclampsia |
| | n = 15 | n = 35 | n = 29 |
| Age (Years) | 32.9 | 28.6 | 27.8 |
| Mean (95% CI) | (29.5 to 36.2) | (22.4 to 34.8) | (26.3–29.3) |
| Parity | 2 | 1 (0 to 6) | 1 (0.5–2) |
| Gestational age (Weeks) | 34.3 | 33 | 37.4 |
| Mean (95% CI) | (32.2 to 36.3) | (24 to 40) | (36.7 to 38.1) |
| BMI (95% CI) | 24.5 (23.0 to 26.0) | 27.0 (19.1 to 34.9) | 27.0 (25.7 to 28.3) |
| Birth weight -grams (95% CI) | 1886.7 (1415.9 to 2357.4) | 1910.9 (997.8 to 2824) | 2600 (2563.6 to 2636.4) |
| Ante-partum IAP -mmHg (95% CI) | 12.5 (10.0 to 15.0) | 13.5 (11.5 to 15.4) | 14.9 (14.6 to15.2) |
| Post-partum IAP mmHg (95% CI) | 9.2 (7.3 to 11.2) | 10.00 (7.6 to 12.4) | 10.2 (9.9 to 10.5) |

IAP- Intra abdominal pressure BMI–Body Mass Index

hypertensive group. Association between IAP and HIP, can be better explained with serial estimation of antepartum IAP in different gestations using a larger sample.

Reduction of IAP from antepartum period to postpartum period is 1.013 times less in HIP, compared to normotensive pregnancy when adjusted for birthweight.

Postpartum IAP level is 2.5 times higher in HIP compared to normotensive pregnancies after adjusting for birth weight, BMI and parity. It is statistically significant.

Our study revealed higher postpartum IAP level and less reduction of IAP level from antepartum period to postpartum period in HIP. It infers the association of high postpartum intra-abdominal pressure and development of postpartum preeclampsia, eclampsia and de novo postpartum hypertension.

While our cross-sectional study provided valuable insights, a prospective cohort study design with serial measurements of IAP in different gestations and postpartum, as well as monitoring maternal and fetal weight gain in HIP and normotensive groups, would offer even more informative data.

## Clinical implications

It is very interesting to consider that preeclampsia only occurs spontaneously in humans and apes, and has been reported in a baboon with twins.–it is a very human and primate specific disease [15]. This may be because almost all other animals carry their young standing on four legs with the abdomen suspended inferiorly, which would be associated with a low IAP in pregnancy.

Exploring the potential therapeutic implications of reducing IAP during pregnancy on HIP remains an understudied area. Interestingly, 60 years ago, a method involving the application of negative abdominal pressure externally was described as a treatment for preeclampsia. A pilot study reported a positive response to this treatment in 90% of cases [16]. However, this device never gained widespread acceptance or use.

Recent study found that prone position may reduce systolic blood pressure in women with preeclampsia without immediate obvious adverse effects [17]. We would suggest prone position or semi prone position in pregnancy may reduce the IAP and development and progression of preeclampsia.

## Conclusion

In normotensive pregnancies at term, mean antepartum IAP was 12.7mmHg (95%CI 11.6 to 13.8) and postpartum mean IAP was 7.3 mmHg (95% CI 6.5 to 8.0). In normotensive pregnancy at term, IAP is within the range of IAH of non-pregnant population.

Higher antepartum and postpartum IAP level, and less reduction of IAP after delivery of the fetus and placenta is associated HIP compared to the normotensive pregnancy.

## Acknowledgments

We would like to thank all volunteers for participation in this study. We thank Kosala Gayan Weerakoon for helpful discussions and support, and Kyriakos ouvelakis for assistance with statistical analysis.

## Author Contributions

**Conceptualization:** Sajith Jayasundara.

**Data curation:** Sajith Jayasundara, Lanka Dassanayake.

**Formal analysis:** Sajith Jayasundara, Malik Goonewardene, Lanka Dassanayake.

**Investigation:** Sajith Jayasundara, Malik Goonewardene, Lanka Dassanayake.

**Methodology:** Sajith Jayasundara, Malik Goonewardene, Lanka Dassanayake.

**Project administration:** Sajith Jayasundara, Malik Goonewardene.

**Resources:** Sajith Jayasundara, Malik Goonewardene.

**Supervision:** Malik Goonewardene, Lanka Dassanayake.

**Writing – original draft:** Sajith Jayasundara, Malik Goonewardene, Lanka Dassanayake.

**Writing – review & editing:** Sajith Jayasundara, Malik Goonewardene, Lanka Dassanayake.

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
