## [Decision Letter · Decision Letter 0]

21 Jun 2023

PONE-D-23-06569The association between maternal intra-abdominal pressure and hypertension in pregnancyPLOS ONE

Dear Dr. Sajith,

Thank you for submitting your manuscript to PLOS ONE. After careful consideration, we feel that it has merit but does not fully meet PLOS ONE’s publication criteria as it currently stands. Therefore, we invite you to submit a revised version of the manuscript that addresses the points raised during the review process.

We look forward to receiving your revised manuscript.

Kind regards,

Surangi Jayakody, MBBS, MSc, MD

Academic Editor

PLOS ONE

Journal Requirements:

Reviewers' comments:

Reviewer's Responses to Questions

**Comments to the Author**

1. Is the manuscript technically sound, and do the data support the conclusions?

Reviewer #1: Partly

Reviewer #2: Partly

2. Has the statistical analysis been performed appropriately and rigorously? 

Reviewer #1: Yes

Reviewer #2: Yes

3. Have the authors made all data underlying the findings in their manuscript fully available?

Reviewer #1: Yes

Reviewer #2: Yes

4. Is the manuscript presented in an intelligible fashion and written in standard English?

Reviewer #1: Yes

Reviewer #2: Yes

5. Review Comments to the Author

Reviewer #1: Thank you for your efforts and hard work.

Can you please clarify these points:

1- Why you only included the patients who are having Caesarian sections? Have you put into your consideration that the pain, the reduced bowel movements, and even the ileus after the C.S can make some bias in measuring the post-operative intraabdominal pressure measurements?

2- Your assumption that the increased intraabdominal pressure is associated with PIH doesn't explain why molar pregnancy is a risk factor for PIH and preeclampsia. As well doesn't explain why we have postpartum preeclampsia and postpartum high BP while the IAP is decreased. on the other hand doesn't explain the normotensive patients who have a reason to have an increase on their IAP as in LGA, Uterine fibroids, and polyhydramnios. can you highlight these conditions, please?

3- What is your objective after concluding that hypertension during pregnancy is associated with raised antepartum and postpartum IAP, any recommendations?

4- All your references are of more than 5 years, Any chance you can have updated references to your topic?

Reviewer #2: Very interesting study, Thank you for bringing new interesting thought of parameter for evaluate Pregnancy induced hypertension.

In Pregnancy induced hypertension one of the main physiological change in maternal body is fluid retention. Therefore I think it is important to find out association between maternal weight gain with Intra abdominal pressure. There could be increase abdominal pressure due to abdominal wall swelling! Also according to study result there is no correlation with fetal weight and abdominal hypertension in Pregnancy induced hypertension group.

I think post partum pain management is very important in planning to measure abdominal pressure in both groups. Therefore it is very important to describe post partum pain management in study.

Thank you!

6. PLOS authors have the option to publish the peer review history of their article (what does this mean?). If published, this will include your full peer review and any attached files.

Reviewer #1: **Yes: **Mena Abdalla

Reviewer #2: No

---

## [Author Response · Author response to Decision Letter 0]

19 Aug 2023

Reviewer #1: 

1- Why you only included the patients who are having caesarean deliveries ? 

IAP measurement includes the insertion of a urinary catheter for IVP measurement. This procedure is commonly done during CD, and the catheter is kept in place until the patient can start moving in the post-operative period. We have decided to utilize this routine practice as an opportunity for our study methodology, rather than introducing an additional intervention for participants. However, for vaginal deliveries, urinary catheter insertion is not a routine protocol. Therefore, we have not included vaginal delivery as part of our study.

This information is now included in the manuscript under Methods- Line number 127-128

2. Have you put into your consideration that the pain, the reduced bowel movements, and even the ileus after the C.S can make some bias in measuring the post-operative intraabdominal pressure measurements?

As part of the post-CD analgesic protocol. To ensure effective pain management, post-operative pain levels are consistently assessed using a pain score, and analgesics are administered following the WHO analgesics ladder guidelines. 

Clinical assessments post-CD focus on identifying potential complications such as post-operative bleeding, surgical infections, constipation, and post-operative ileus. None of the participants experienced any of these complications during their hospital stay following the operation.

This information is now included in the manuscript under Methods - Line number 141-144

3 . Your assumption that the increased intraabdominal pressure is associated with PIH doesn't explain why molar pregnancy is a risk factor for PIH and preeclampsia. As well doesn't explain why we have postpartum preeclampsia and postpartum high BP while the IAP is decreased. on the other hand doesn't explain the normotensive patients who have a reason to have an increase on their IAP as in LGA, Uterine fibroids, and polyhydramnios. can you highlight these conditions, please?

The development of PIH is influenced by multiple factors. The placenta plays central role in the pathogenesis of preeclampsia, specifically in relation to the imbalance of placental pro-angiogenic and anti-angiogenic factors. Genetic factors and immune factors have also been established as contributors to the development of preeclampsia. Additionally, increased IAP may serve as a contributing factor [1]. The extent of each factor's contribution to the pathogenesis may vary depending on individual circumstances.

In the pathogenesis of preeclampsia, the presence of amplified anti-angiogenic factors from the placenta is believed to be particularly prominent in cases of molar pregnancy as compared to other factors. Additionally, changing partners between pregnancies increases the risk of preeclampsia, where immune maladaptation at the fetal-maternal interface could be an underlying mechanism more than other factors.

The role of IAP in the development of preeclampsia may align with the traditional concept of disease causation, where the virulence of the agent and the susceptibility of the host play a crucial role. Consequently, elevated IAP might not always lead to preeclampsia in situations that raise the IAP, for instance, polyhydramnios, fetal macrosomia, uterine fibroids, or ovarian cysts. 

After delivery of fetus and placenta, IAP levels typically decrease. Nevertheless, the effects of pathological processes following delivery can either linger or diminish, contingent upon individual factors. In particular, the presence of elevated levels of placental pathological agents in circulation post-delivery can contribute to the development of postpartum preeclampsia and hypertension [2].

1. Marshalov DV, Shifman EM, Salov IA, Petrenko AP, Ioscovich A. Preeclampsia is a Syndrome of Intra-Abdominal Hypertension in Pregnancy - would a Hypothesis become a Theory?. J Clin Anesth Manag. 2017;2(1): doi http://doi.org/10.16966/2470-9956.122

2. Brien ME, Boufaied I, Soglio DD, Rey E, Leduc L, Girard S. Distinct inflammatory profile in preeclampsia and postpartum preeclampsia reveal unique mechanisms. Biol Reprod. 2019 Jan 1;100(1):187-194. doi: 10.1093/biolre/ioy164. PMID: 30010720

This information is now included in the manuscript under Introduction- Line number 92- 109

4. What is your objective after concluding that hypertension during pregnancy is associated with raised antepartum and postpartum IAP, any recommendations?

It is very interesting to consider that preeclampsia only occurs spontaneously in humans and apes, and has been reported in a baboon with twins. – it is a very human and primate specific disease [1]. This may be because almost all other animals carry their young standing on four legs with the abdomen suspended inferiorly, which would be associated with a low IAP in pregnancy.

Exploring the potential therapeutic implications of reducing IAP during pregnancy on PIH remains an understudied area. Interestingly, 60 years ago, a method involving the application of negative abdominal pressure externally was described as a treatment for preeclampsia. A pilot study reported a positive response to this treatment in 90% of cases [2]. However, this device never gained widespread acceptance or use. 

Recent study found that prone position may reduce systolic blood pressure in women with preeclampsia without immediate obvious adverse effects [3]. We would suggest prone position or all four-leg position in pregnancy may reduce the IAP and development and progression of preeclampsia. 

1. Chau K, Welsh M, Makris A, Hennessy A. Progress in preeclampsia: the contribution of animal models. J Hum Hypertens. 2022 Aug;36(8):705-710. doi: 10.1038/s41371-021-00637-x. Epub 2021 Nov 26. PMID: 34837033; PMCID: PMC8617007.

2. Blecher JA, Heyns OS. Abdominal decompression in the treatment of the toxaemias of pregnancy. Lancet. 1967;2(7517):621-624. doi:10.1016/s0140-6736(67)90678-2

3. Dennis AT, Hardy L, Leeton L. The prone position in healthy pregnant women and in women with preeclampsia - a pilot study. BMC Pregnancy Childbirth. 2018;18(1):445. Published 2018 Nov 16. doi:10.1186/s12884-018-2073-x

This information is now included in the manuscript under Discussion- Line number 259- 272

5. All your references are of more than 5 years, Any chance you can have updated references to your topic?

Agreed with your valuable comment. There is a dearth of scientific literature on this topic. However, we have included recently published evidence. We included following references:

1. Arora V, Tyagi A, Ramanujam M, Luthra A. Intraabdominal pressure in non-laboring preeclamptic vs normotensive patients undergoing cesarean section: A prospective observational study. Acta Obstet Gynecol Scand. 2020; 99: 1031- 1038.

2. Arruda Correia ML, Peixoto Filho FM, Gomes Júnior SC, Peixoto MVM. Effects of intra-abdominal hypertension on maternal-fetal outcomes in term pregnant women: A systematic review. PLoS One. 2023;18(6):e0280869. Published 2023 Jun 27. doi:10.1371/journal.pone.0280869

This information is now included in the manuscript under Discussion- Line number 229- 231 and 237 -242

Reviewer #2: 

1. In Pregnancy induced hypertension one of the main physiological change in maternal body is fluid retention. Therefore I think it is important to find out association between maternal weight gain with Intra abdominal pressure. There could be increase abdominal pressure due to abdominal wall swelling! Also according to study result there is no correlation with fetal weight and abdominal hypertension in Pregnancy induced hypertension group.

Agreed with your valuable comment. We conducted a cross sectional study. However, prospective cohort study design with serial measurement of IAP in different gestations and postpartum, maternal weight gain, fetal weight gain in HIP and normotensive groups may be more informative. 

In our study, IAP is correlate with birth weight in normotensive pregnancy. Interestingly, It is not correlate with birthweight in HIP group. It raises the suspicious of “IAP in PIH group may not only result of growing fetus compare to the normotensive group – Additional increased IAP may play role in pathogenesis of HIP’’. 

2. I think post partum pain management is very important in planning to measure abdominal pressure in both groups. Therefore it is very important to describe post partum pain management in study.

As part of the post-CD analgesic protocol. To ensure effective pain management, post-operative pain levels are consistently assessed using a pain score, and analgesics are administered following the WHO analgesics ladder guidelines. 

Clinical assessments post-CD focus on identifying potential complications such as post-operative bleeding, surgical infections, constipation, and post-operative ileus. None of the participants experienced any of these complications during their hospital stay following the operation.

This information is now included in the manuscript under Methods - Line number 141-144

---

## [Decision Letter · Decision Letter 1]

3 Oct 2023

The association between maternal intra-abdominal pressure and hypertension in pregnancy

PONE-D-23-06569R1

Dear Dr. Sajith,

We’re pleased to inform you that your manuscript has been judged scientifically suitable for publication and will be formally accepted for publication once it meets all outstanding technical requirements.

Kind regards,

Surangi Jayakody, MBBS, MSc, MD

Academic Editor

PLOS ONE

Additional Editor Comments (optional):

Reviewers' comments:

Reviewer's Responses to Questions

**Comments to the Author**

1. If the authors have adequately addressed your comments raised in a previous round of review and you feel that this manuscript is now acceptable for publication, you may indicate that here to bypass the “Comments to the Author” section, enter your conflict of interest statement in the “Confidential to Editor” section, and submit your "Accept" recommendation.

Reviewer #1: All comments have been addressed

Reviewer #3: All comments have been addressed

2. Is the manuscript technically sound, and do the data support the conclusions?

Reviewer #1: Yes

Reviewer #3: Yes

3. Has the statistical analysis been performed appropriately and rigorously? 

Reviewer #1: Yes

Reviewer #3: I Don't Know

4. Have the authors made all data underlying the findings in their manuscript fully available?

Reviewer #1: Yes

Reviewer #3: Yes

5. Is the manuscript presented in an intelligible fashion and written in standard English?

Reviewer #1: Yes

Reviewer #3: Yes

6. Review Comments to the Author

Reviewer #1: Thank you for addressing all the previous reviewers' comments. I would accept this paper so far for publication.

Reviewer #3: Congratulations to the authors who have addressed a rarely explored topic. Preeclampsia is an important topic and the current article notes an association of IAP around the time of delivery with HIP. The article is well written and an interesting read. Previous reviewer comments have all been addressed in the current version. Recommendation- Accept

7. PLOS authors have the option to publish the peer review history of their article (what does this mean?). If published, this will include your full peer review and any attached files.

Reviewer #1: **Yes: **Mena Abdalla

Reviewer #3: **Yes: **indu asanka jayawardane

---

## [Editor Report · Acceptance letter]

9 Oct 2023

PONE-D-23-06569R1 

The association between maternal intra-abdominal pressure and hypertension in pregnancy 

Dear Dr. Jayasundara:

I'm pleased to inform you that your manuscript has been deemed suitable for publication in PLOS ONE. Congratulations! Your manuscript is now with our production department. 

Kind regards, 

on behalf of

Dr Surangi Jayakody 

Academic Editor

PLOS ONE